# Children’s Experiences of Their Journey to School: Integrating Behaviour Change Frameworks to Inform the Role of the Built Environment in Active School Travel Promotion

**DOI:** 10.3390/ijerph18094992

**Published:** 2021-05-08

**Authors:** Nafsika Michail, Ayse Ozbil, Rosie Parnell, Stephanie Wilkie

**Affiliations:** 1Department of Architecture & Built Environment, Northumbria University, Newcastle upon Tyne NE1 8ST, UK; ayse.torun@northumbria.ac.uk; 2School of Architecture, Planning and Landscape, Newcastle University, Newcastle upon Tyne NE1 7RU, UK; rosie.parnell@newcastle.ac.uk; 3School of Psychology, University of Sunderland, Sunderland SR1 3SD, UK; stephanie.wilkie@sunderland.ac.uk

**Keywords:** active commuting to school, child-centred approach, children’s attitudes, built environment, behaviour change, health promotion

## Abstract

Childhood obesity is a public health problem with multiple effects on children’s life. Promoting Active School Travel (AST) could provide an inclusive opportunity for physical activity and shape healthy behaviours. Data for this cross-sectional study were drawn from questionnaires carried out in five primary schools located in Newcastle upon Tyne, UK, in neighbourhoods chosen for their variability in IMD (index of multiple deprivation) and spatial structure of street networks (measured through space syntax measure of integration). A randomly selected and heterogenic sample of 145 pupils (aged 9–10) completed an open-ended questionnaire to state what they like and dislike about their journey to school. Thematic analysis identified four typologies (environmental context, emotions, social influences and trip factors) based on the Theoretical Domains Framework (TDF) and specific themes and sub-themes underlying children’s affective experiences of their journeys to school. This study is the first known to authors to attempt to adapt the Capability, Opportunity and Motivation Behaviour (COM-B) model into AST and children’s experiences and associated behavioural domains with design aspects. Such an insight into children’s attitudes could inform urban planners and designers about how to apply more effective behaviour change interventions, targeting an AST increase among children.

## 1. Introduction

In the context of global urbanisation, every year more children are born and raised in urban environments, with 70% of children and youth projected to live in cities by 2050 [1]. The way urban forms are structured could influence human health and ways of living [2]. Extensive research indicates that the built environment can have an impact on population health, affecting diabetes and obesity [3], respiratory diseases [4] and cardiovascular diseases [5]. At the same time, built environments can have a preventative role in human health by promoting healthy behaviours, such as physical activity [6], healthy diet [7] and opportunities for social cohesion and restoration [8]. Despite the importance of the built environment for children’s health and development [2], including potential effects on obesity and metabolic syndrome in adult life [9], less evidence exists on the influences of the built environment on children’s health behaviours. More specifically, the role of the built environment in promoting healthy behaviours has been reviewed as part of the obesogenic environments [10]; although there are some urban planning initiatives to define the design principles of child-friendly cities, such as community spaces, green infrastructures that promote playing and traffic measures to prioritise pedestrians and cyclists [11,12,13], connections between urban forms and promotion of children’s active behaviours are not yet clear.

According to World Health Organisation guidelines, children and adolescents aged 5–17 should do at least an average of 60 min per day of moderate- to vigorous-intensity, mostly aerobic, physical activity across the week, in order to improve physical fitness (cardiorespiratory and muscular fitness), cardiometabolic health (blood pressure, dyslipidaemia, glucose and insulin resistance), bone health, cognitive outcomes (academic performance, executive function) and mental health (reduced symptoms of depression) and reduce adiposity ([14] (p. 1)). Physical activity typically refers to sporting and fitness activities; physical education; active play and informal activity; and transportation, including walking, cycling and wheeling [14,15]. Active school travel (AST), both as a direct means of physical activity and as an opportunity for active play and informal activity, could be considered as a public health intervention to promote physical activity among children. AST can potentially provide an inclusive opportunity for physical activity [16] and shape life-long healthy behaviours [17]. As a means of physical activity, active travelling can benefit children’s physical health [18,19] and cognitive development [20]. Simultaneously, AST also provides an opportunity for socialising [21] and exploration of neighbourhood localities, which help create a child’s identity as part of the community [22]. Similarly, AST is found to give a sense of freedom and independence to children [23], a characteristic that overlaps with the use of public spaces and parks [24]. Previous research also indicates more children would walk or cycle to school if they had the choice [25] and that children would like to have more opportunities for physical activity and engagement within open public spaces [24].

Everyday travel creates complex behaviour patterns due to modern ways of living and existing urban forms [26]. This means that promoting AST is a complex public health challenge. Previous AST models and frameworks suggest a complex relationship and dynamic interaction between individual characteristics and household, social and environmental influences on the decision-making process of school travel [27,28,29]. The M-CAT model suggests that apart from the household, socio-economic and cultural characteristics, the natural and built environment, as well as the perceptions of them, affect the choice of active travelling to school [28]. The Behavioural Model of School Transportation defines the influences of AST through five main domains: proximity, traffic and personal safety, street connectivity, comfort and attractiveness and opportunity for social capital [29]. Previous studies indicate that proximity to school is the most crucial factor [16,30,31,32,33], and distance was consistently found to be negatively associated with AST [34]. Traffic-related walking safety is also a strong determinant of AST [35,36], with speed limits and crossing aids in school neighbourhoods associated with more active trips [37]. On the other hand, street connectivity may support AST [32,38] but also may encourage more car traffic [39]. Attractiveness and aesthetics, including well-maintained green spaces and absence of litter, have been shown to be associated with older children’s (often independent) trips [36], although findings are inconsistent [29]. Similarly, land-use mix may be associated with shortest destinations and therefore better accessibility [32], although there is a lack of clear evidence on how the variety of land uses may support AST. Weather and topography may also affect active travelling [29], but this varies with cultural context [38]. Nonetheless, perception of school routes rather than objective qualities of the built environment may be more crucial for AST decision-making [35]. Personal safety and sense of community are found to be related to parents’ perceptions rather than objective measures of the built environment [32], which in turn may influence children’s independent mobility levels [40].

Nevertheless, children’s perceptions of the built environment may be significantly different from those of their parents [41], and children’s preferences can influence their parents’ decisions [28]. Therefore, it is important to understand children’s opinions in this context. According to previous participatory research with children [42,43,44], their views and local knowledge can help better understand how the built environment affects school travel behaviour and contribute to giving context and value to interventions that focus on their health and well-being [43]. Behavioural science would support this, as AST could be either reasoned (i.e., walking to school because it is the best possible option) or habitual (i.e., walking to school as part of a morning routine) behaviour [45], and therefore, attitudes are a crucial part of exploring school travel. 

Attitudes are related to positive or negative views that people have regarding any aspect of reality [46]; according to the ABC model, attitudes consist of cognitive, affective and behavioural components [47]. Previous research on adults found that all three play an important role in explaining travel-related decisions [48], with fewer studies on children’s affective attitudes. The school journey is found to be the least pleasant affective travel experience for children [49], while studies show that in addition to the physical and social environments, activities that take place along the journey to school and the means of travelling are themselves important aspects of children’s affective experiences. Despite this, children’s school travel preferences are scarcely explored in the literature [43,44].

Overall, school travel is a behaviour and, like many other health-related behaviours, is difficult to change [50] and maintain [28]. At the same time, there is an opportunity for planners and designers, many of whom are unfamiliar with behaviour change frameworks, to support a behaviour change through design. A multidisciplinary understanding of the factors and motivations [51] that influence children’s travel behaviour is needed in order to design and apply effective transport policies [29]. Environmental psychology suggests that perception and subjective evaluation of the built environment may influence travel behaviour [52]; however, the bridge between physical environment and behavioural science is still unclear [53]. Behavioural science frameworks offer a means to explore the environmental factors that facilitate supportive or hindering experiences for children, allowing connections to be made between urban design and AST behaviour change. The Capability, Opportunity and Motivation Behaviour model (COM-B) and the Theoretical Domains Framework (TDF) are two of the behaviour change frameworks that can be used to explore travel behaviour since they both include attitude and cognate aspects of behaviours and the role of the built environment in promoting physical activity among adults [53]. The TDF is a summary of the main domains that influence humans’ behaviour [54], including environmental context and resources, social influences and emotions (Figure 1a). COM-B is a behaviour change model currently used in different types of behaviour change interventions and understands human behaviour to be a result of physical and psychological capability, physical and social opportunity and the automatic (emotional) and reflective (rationale) motivation to the varied influences on decision-making [55], (Figure 1b). The TDF has been used previously to explore parents’ perception of AST [40], but not children’s perception of their journey to school. Both TDF and COM-B provide an opportunity for analysis of the behavioural influences of AST and, in contrast with other behavioural models, could contribute to designing AST interventions focusing on components of the built environment. Taking into consideration that a supportive built environment could result in behaviour changes for children [27], it is proposed that exploring their experiences and preferences through behaviour change models can help understand children’s preferences and the underpinning reasons for their travel behaviours that may help to develop guidelines for urban designers and planners, towards supporting more active trips to school.

Consequently, the aim of this paper is to explore children’s affective experiences of their journey to school, using the TDF and the COM-B as frameworks to analyse children’s attitudes and inform the role of the built environment in AST promotion. Such an approach addresses the above gaps in the literature and makes an important contribution to the limited knowledge related to children’s affective experiences during school travel, by offering the novel approach of bridging built environment attributes with behaviour change frameworks.

## 2. Materials and Methods

### 2.1. Location and School Selection

Newcastle upon Tyne is the largest city in the North East of England with a population of 302,820 [56] people, and it is located on the northern bank of the River Tyne, covering an area of approximately 115 km^2^. According to the Newcastle Children and Young People’s Health and Wellbeing Survey 2019, only 39% of primary school children walked to school and 6% used scooters and bikes [57], while the childhood obesity rates in Newcastle upon Tyne are significantly higher than the England average and above the North East average [58]. In addition, 32% of boys and 35% of girls in Newcastle upon Tyne are estimated to exercise outside school only once per week, whilst 13% had no opportunity for any physical activity [57]. For the above reasons, Newcastle upon Tyne was selected as a case study to explore children’s school journey to school and built environment attributes.

After geocoding all primary schools in Newcastle upon Tyne, schools were classified into four different groups, according to a matrix of street connectivity and socio-economic context (Figure 2). Street connectivity was measured using space syntax measure of 2 km integration based on Space Syntax Open Mapping [59]. Integration measures how accessible each space (street segment) is from all the others within the radius using the least angle measure of distance. Socio-economic context was measured based on open source CDRC index of multiple deprivation [60]. Schools in suburban parts of the city were excluded to ensure a similar level of urbanisation [39]. An invitation to participate in the study was sent through an email to head-teachers of the selected schools, according to the four groups. Four schools in physically and socially heterogeneous neighbourhoods, according to the above classification, replied to the invitation, while one extra school participated as a pilot study. Participating schools’ interest to participate in this research project, which aimed to investigate how urban design and planning could promote healthy behaviours and physical activity among children, made the data collection possible.

### 2.2. Sampling

Although the sampling of schools was cross-sectional, the sampling of participants was purposive, targeting a range of experiences of children’s journeys to school. All students in year 5 (9–10 years old) at these schools (~40 students per school, *n* = 192) were invited to participate without any exclusive criteria. Relevant consents were received for 145 participants. Ethical approval to conduct the study was received by the host institution ethics committees (30 April 2019, Submission Ref: 15592). Table 1 shows the participants’ characteristics. Out of the 145 participants, 54% were girls and 45% were boys. Half of the pupils typically walked to school, while 30% commuted to school by car. Only 5% used a bike to travel to school, 3% participated in a park and stride programme, 3% used public transportation (bus) and 5% used the school bus. All of those who cycled to school were living in the same neighbourhood, with cycling infrastructure, and all children who participated in a park and stride programme were from the same school. Out of the 145 participants, 15 (10%) were travelling without an adult to accompany them and 5 (3%) mentioned that they sometimes travelled alone. All the participants that travelled alone or with other child(ren) either walked or cycled to school. Finally, a similar percentage of those who walked and those who commuted by car sometimes stopped somewhere else, as part of their journey to school, which was usually a shop.

The pilot study was conducted in May 2019, and the main study was conducted during November 2019–March 2020. Participatory mapping activities were carried out with the participating children at each of the five schools. This paper focuses specifically on the methods and data relating to children’s affective experiences of their journey to school. On a day and time agreed with the teacher, the researcher brought an A3 leaflet for each student to complete, which included a child-friendly designed map [61,62], a closed question regarding children’s usual mode of travel, open-ended questions related to their experiences of their journeys (if they ‘travel with or meet’ somebody and if they ‘stop’ in any particular place during their journey to/from school) and open-ended questions about what they liked or disliked about their journey to/from school [63]. The activity took place in the classroom for approximately 30 min, and children were invited to answer any of the questions that were relevant to their experience, to use either words or drawings to express themselves [63] and to leave blank any of the questions they did not wish to answer [61]. The researcher, the teacher and teaching assistants were present. Questions were clarified where appropriate, but no guidance was offered to children about what answers to give [43].

Students’ short open-ended responses were organised into two lists: what children like and what they dislike about their journey to school, retaining a table with the ID of each student and all descriptive characteristics: ‘name’, ‘school’, ‘mode of travel’, ‘travel with’ and ‘stop’. As far as mode of travel was concerned, ‘walking’ and ‘cycling/scootering’ were considered as separate active travel modes, due to the different experiences they may provide [48]. 

### 2.3. Analysis

After getting familiarised with children’s comments [64], a qualitative thematic analysis was used to identify patterns [65] among participants’ open-ended responses. Once an initial framework of themes and sub-themes [64,65] was created, the table with participants’ characteristics and comments was imported into NVivo 12 (QSR International Pty Ltd. Version 12, 2018). All descriptive characteristics (e.g., school, gender, commuting mode, travel with, stop) were imported as closed responses, while children’s likes and dislikes were imported as open-ended responses and coded into nodes, under the main nodes of ‘likes’ and ‘dislikes’. The thematic analysis was both deductive, using themes already reported in the existing literature, and inductive, with themes emerging from children’s answers [66] (Figure 3). However, as the scope of this paper is to explore children’s experiences according to behaviour change frameworks, themes and sub-themes were then grouped into typologies (wider themes) [65], based on TDF (Figure 3), in order to explore them through the COM-B model [55,67] (Figure 4). Finally, the nodes were visualised as charts, coding by ‘attribute value’ NVivo analysis option, to explore any associations between themes and schools, between themes and genders and between themes and means of travelling to school [43].

## 3. Results

A small percentage of children replied ‘nothing’, ‘anything’ or ‘everything’ to what they like or dislike about their journey to school, while others left one or both sections blank, even though they completed other open-ended questions and/or returned the leaflet to the researcher. It is important to notice that those responses or the lack of reply was among children who either actively walked or travelled by a car. The rest of children’s replies are presented according to the thematic analysis and based on the four behavioural domains. 

### 3.1. Environmental Context

The aspects of environmental context, as raised in children’s comments, can be grouped into physical and natural environment. Physical environment was a popular theme emerging from children’s comments, including both ‘likes’ and ‘dislikes’. Children often referred to aspects of the physical environment that directly influence their experience of everyday commuting to school. Environmental context was common to be considered as a positive influence on children’s travel to school experiences, with comments referring to environmental attributes to be positive aspects of children’s journey to school.

One of the sub-themes strongly represented in the children’s comments was the distance between school and home or time of travel between school and home. Therefore, the proximity to school was highlighted both in positive and negative comments: ‘close to home’, ‘takes 2 min to get to school on foot’, ‘long way’. Apart from proximity to school, other neighbourhood design features were also mentioned by participants as liked or disliked environmental attributes. The ‘variety of ways to go to school’ and ‘quiet alleys to walk and cycle’, which are related to street connectivity and route options, along with ‘new buildings’ and ‘beautiful places’, were also some of the aspects children referred to in their like section, while ‘lack of fun places’, ‘too many streets’ or ‘seeing the same landscape every day’ were part of the dislikes. Land-uses was a common theme within the environmental context. The existence of shops, especially ‘sweet shops’ and the ‘smell’ of food shops, were popular among children’s comments, while ‘seeing’ and ‘going through’ parks was another sub-theme related to land-uses. Street design was also a recurring theme in children’s responses. ‘Seeing front gardens’ emerged as a positive aspect of their journey to school, while the ‘lack of trees’ was a negative aspect. Similarly, ‘lack of zebra crossings’ and ‘no traffic lights’, ‘existence of ‘bumpy roads’, ‘puddles’, ‘works’ or ‘ice on the pavement’ and ‘muddy paths’ were disturbing features along their journey to school. In addition, children referred to ‘cycling paths’, both in their likes and dislikes sections. 

Besides the built environment, children also mentioned features of the natural environment, including weather, wildlife and the morphology of terrain, with uphill routes, in both likes and dislikes sections. Fresh air was a popular theme among children’s responses and categorised within the typology of environmental context in the current study; this theme of fresh air was not one previously highlighted in other studies. ‘Smelling flowers’, ‘seeing plants’ and ‘picking flowers’, ‘seeing birds’ and ‘listening to birds’ or even ‘walking on leaves’ were some further sub-themes children mentioned as liked features within the theme of nature. On the other hand, the lack of habitats and existence of dead animals and cutting of trees in public spaces were referred to as dislikes: ‘not many habitats for animals’, ‘deforestation’. 

It is interesting to note that similar aspects of neighbourhood design were mentioned by different children residing in similar environmental contexts. For example, topography was mentioned as a positive sub-theme only in a particular school in an uphill and downhill area and as a negative aspect only in another school, where the neighbourhood is relatively flat. Similarly, cycling paths emerged as a sub-theme, both positive and negative, only among children of two schools located in the same neighbourhood where cycling paths exist. On the other hand, proximity was not mentioned as a positive aspect for children attending the school located in a neighbourhood with a relatively dense street network. Nevertheless, aspects of the physical environment related to street connectivity, as mentioned above, were mentioned only in the positive comments of children residing in a neighbourhood with relatively high street connectivity (regular grid-iron layout and relatively smaller blocks). 

### 3.2. Emotions

A second domain emerging from children’s responses was their emotions related to perceived environmental and personal safety in the public space. Children’s comments in this typology were commonly mentioned in the dislike section of children’s journey to school. Some of these were associated with traffic-related and personal safety, and others are related to comfort and attractiveness. 

Children indicated that they felt unsafe due to ‘car traffic’ and ‘speeding cars’, and they mentioned ‘crossing the street’ and ‘parked cars at street corners’ as aspects of discomfort during their journey to school. ‘Noise’ and ‘air pollution’ were also popular sub-themes within traffic-related themes in the dislike section. As expected, children who walked regularly to school usually referred to ‘crossing’ as a sub-theme; however, in one particular school, neither ‘crossing’ nor ‘car traffic’ was mentioned by children commuting with either an active or a non-active way. 

Strangers, such as ‘nasty people’ and the ‘fear of being abducted’, were mentioned as a disliked part of the journey as well. Other features related to discomfort due to social interaction in public spaces included ‘bullies blocking the path’, ‘bikers hooting on the cycle path’ and ‘people stepping into cycle path’. Similarly, incivilities in public spaces related to other people’s behaviours emerged as features causing discomfort among children, ‘people shouting’ and ‘smoking’ as well as ‘litter’, ‘dogs’ poo’ and ‘broken glass’, as well as ’cars splashing water’, ‘crowded paths’ and ‘walking with others’. On the other hand, feeling ‘safe’ and ‘calm’ were mentioned by children as positive emotions along their routes to school.

### 3.3. Social Influence

Children’s interaction with their families, friends and the larger community emerged as a third domain that influenced their experiences along their journey to school. Similar to environmental context, social influence was primarily considered positively by children. It is interesting to notice that this typology was particularly prevalent among children who typically walked to school.

‘Talking’ and ‘playing with parent(s)’ along the trip to school was a strongly represented theme by children. Similarly, ‘meet’, ‘talk’ and ‘play with friends’ and sometimes with siblings was a common theme that pupils made comments about. Another family-related sub-theme was ‘walking dogs’ along the journey to school. However, children also referred in a negative way to their interactions with family members and friends. For example, ‘sibling walking in front of me’, ‘parent walking too fast’ or ‘friends bullying’ were some of the comments. Another aspect emerging in the dislike section was ‘saying goodbye’ to parents or pets.

Apart from family members and friends, children reported ‘grandparent’s’, ‘cousin’s’ and ‘friend’s house’ along the journey as a positive aspect of their trip to school. ‘Saying hello to neighbours’ and ‘meeting dogs’ along the journey were also mentioned within the positive responses. Additionally, comments related to school as a destination and place to meet friends were raised in the like section: ‘seeing school people’, ‘meet my friends at school’.

### 3.4. Trip Factors

The final typology included individual circumstances related to methods of commuting or consequences of the commute itself. For example, students commented on walking/cycling/scootering/bus ride/being in the car as both positive and negative aspects of their journey.

Participants commented on the way of travelling and parts of the journey itself. ‘Walking’, ‘cycling’, ‘scootering’, ‘bus ride’ and ‘being in a car’ were mentioned by children as both positive and negative aspects of their journey. On the other hand, children also referred to parts or consequences of the trip itself only in a negative way: ‘fatigue’ or ‘falling’ during walking or ‘uncomfortable seats’ and ‘lack of fresh air’ when in a car. ‘Being late to school’ (e.g., due to late/slow buses) was another sub-theme emerging for non-active modes. The speed and length of the trip were both referred to as positive and negative aspects by children commuting by car: ‘go faster to school’, ‘driving slow’. 

Activities along the journey to school were a strongly represented sub-theme in the children’s comments, within the travel theme. While passive activities such as ‘listening to music’, ‘singing’, ‘looking out of the window’, ‘relaxing’ or ‘sleeping’ were reported as liked activities by those children travelling by car or bus, children actively travelling to school primarily mentioned physical activities such as ‘playing’, ‘jumping’, ‘skipping’, ‘running’ and ‘exercise’ as positive aspects of their journey to school. ‘Dropping siblings to school’ was also mentioned within both the ‘like’ and ‘dislike’ categories. According to our results, children walking to school referred to a bigger variety of sub-themes compared with those mentioned by children using any other commuting mode. 

Some comments related to the school day, such as having ‘no class’ or it ‘being Friday’, were mentioned positively for children’s experiences, and others related to morning routine, such as ‘waking up early’ in the morning or ‘being sleepy’, belonged to the dislike category.

## 4. Discussion

Increasing children’s active travel rates to school is important for both health and environmental reasons. Understanding children’s affective experiences of their journeys to school is an important step in identifying the built environment factors that hinder or support active school travel (AST). This study provided an in-depth exploration of features that underlie children’s affective experiences of school travel through a thematic analysis, taking into consideration different ways of commuting among 9–10-year-old children who live in physically and socially heterogeneous neighbourhoods.

### 4.1. Linking the Results with Behaviour Change Frameworks

Our results are consistent with findings from the previous limited number of studies on children’s experiences of their journey to school [22,43,44,68], with a new theme around air pollution and fresh air emerged from our analysis. Nevertheless, the scope of this article was to explore these findings further in order to bridge the gap between urban planning/design aspects and behaviour change interventions. The built environment is part of the school travel experience, and therefore, developing a conceptual analysis of its role in behaviour change could bring a clearer pathway to urban designers and planners in implementing their interventions in the urban sphere around schools to promote AST and children’s health. The findings of this study can be summarised under four headings based on the TDF: environmental context, emotions, social influences and trip factors [40,67]. Moving further, integrating the COM-B model on those headings could work as a guide [54] to how the built environment could contribute to sustainable behaviour change, offering a framework for how to design a complex behaviour change intervention [69].

#### 4.1.1. Environmental Context as a Physical Opportunity

According to the COM-B model, the environmental context provides the physical opportunity to promote a specific behaviour [68]. Smaller distances and therefore shorter durations of the home–school trip could act as physical opportunities to promote AST, since they are both important features of children’s affective experiences. Previous literature highlights the need for a better policy related to school selection and planning of school locations [70,71]. However, since even the children residing within a relatively short distance of school may be driven to school [16], urban planning and design should also focus on other factors that may affect the journey experience. The results of this study indicate that the built environment is an important component of children’s school journey. Particularly, neighbourhood and street design, including street connectivity and street trees, land-uses, active travel infrastructures (e.g., cycling path) and nature, emerged as significant environmental features underlying children’s affective experiences, most often in a positive way. Active travel infrastructure supports walking and cycling [72], and therefore, planning active travel infrastructure within neighbourhoods around schools to accommodate children’s preferences has the potential to increase active trips to school. Although the majority of children participating in this study were accompanied by adults during their travel to school, and hence their attitudes and behaviours may be influenced by their parents [73], the fact that similar aspects of neighbourhood design are mentioned by different children residing in similar environmental contexts (i.e., active travel infrastructures, street connectivity levels) indicates that objective features of the built environment play a significant role in shaping children’s affective experience of everyday travel.

Among behaviour change techniques [74], ‘restructuring the physical environment’ and ‘adding objects to the environment’ ([75] (p. 702)) are claimed to be two strategies linked to the physical environment. Hence, these can be used to introduce changes to the environmental context as part of a complex behaviour change intervention [75], such as promoting physical activity and AST. Taking into consideration children’s comments regarding the variety of the environmental context (e.g., seeing different buildings, lack of fun places, seeing the same landscape every day) and the opportunity to have a variety of routes to school, integrating a mix of land-uses and fun places along school routes could make children’s travel experience more interesting, which might also lead to positive emotional consequences (i.e., can make children happier, less stressed) [74]. The negative comments regarding the number of streets and the positive comments on the existence of shops, parks and front gardens along the school route that stimulate children’s senses (e.g., smelling food, hearing birds, seeing flowers) may indicate that school routes with multiple stimuli could provide more pleasant experiences and therefore more opportunities for a behaviour change [68]. Similarly, a well-connected street network in school neighbourhoods could provide a variety of routes [76]. In addition, providing routes with fresh air (e.g., less traffic, more trees and pocket parks) may encourage more children to walk to school, while simultaneously contributing to urban nature and ecology [77,78], which in turn could enhance the multisensory experience of a school journey. 

Apart from the physical opportunity that an environmental context could provide for more active trips, the built environment may have an impact on other domains, such as emotions and social influence, which affect motivations and opportunities for a behaviour change. By exploring this impact, urban planning and design could contribute to creating cities, neighbourhoods and school streets supportive of active travel among children. According to the behavioural model of school transportation [29], the urban environment could have an impact on travel decisions and AST. If the aim is the sustainable promotion of AST, understanding the environmental impact on children’s affective experiences is crucial in order to design for more pleasant trips.

#### 4.1.2. Emotions as an Automatic Motivation

Emotions may work as an automatic motivation towards a behaviour change [68], and therefore an experience that is connected with positive emotions may result in a sustainable change and maintenance of a specific behaviour. Similar to Wilson et al. [44], the findings suggest that children’s journeys to school are connected with a variety of emotions. According to these findings, perceptions of neighbourhood safety—both traffic-related (e.g., speeding cars) and fear of strangers (e.g., fear of being abducted)—and environmental incivility (e.g., litter) are important aspects of children’s travel experiences and can affect their emotions, which has been well documented in previous literature [67]. Pocock et al. [35] suggest that besides enabling active travel, safe routes to school can provide positive experiences that may increase sustainable AST in the long term. Carefully designed school streets [73] and green shortcuts [43] can ensure safer, quieter and cleaner travel experiences for children along their journey to school. Similarly, children reported negative experiences of crowded paths or interactions between themselves and pedestrians or cyclists. This suggests that a more effective design of footways and dedicated cycle lanes that enable a better segregation between different groups could provide improved experiences. Designing for pleasant experiences and safer journeys can promote automatic motivation and positive attitudes among children towards active travel, physical activity and engagement with their local neighbourhoods. 

#### 4.1.3. Social Influence as a Social Opportunity

The social opportunities of urban settings are suggested to be considered for any intervention targeting the promotion of physical activity among children [62], and according to Atkins et al. [68], this could be provided by the behaviour domain ‘social influence’. Our results align with previous studies [22,43,44,67], indicating that social interactions with family and friends are a crucial aspect of children’s journeys, regardless of the mode of travel. On the other hand, antisocial behaviours and incivilities cause discomfort to children and could discourage future active trips. Our findings suggest that the interaction with the larger community was regarded as a positive aspect, especially for children walking to school. Hence, in order to enable social opportunities and promote AST among children, the built environment should encourage a cohesive community that accommodates social interactions that pupils enjoy on their journey to school (i.e., meeting/playing/talking with friends and neighbours along the route). This could include policies on density, proximity and school selection that may allow the development of local relationships and trust. In addition, improving the physical quality of streets and public spaces, and hence residents’ environmental perception, could influence social cohesion [79]. 

#### 4.1.4. Trip Factors as a Reflexive Motivation

A behaviour may be driven by an individual’s beliefs, identity and goals which may form a reflexive motivation for the specific behaviour [70]. In the case of AST, parents’ decisions for their children commuting to school may be a result of their social identity and personal priorities [80], which may have an impact on children’s experiences. According to our results, children’s affective experiences may differ depending on how they travel to school. Similar to Egli et al. [43], we found that children actively commuting to school mentioned a wider range of sub-themes, both for physical and social environments, than those who are commuting by car. In addition, our results indicate that different ways of commuting may encourage other types of activities, with children walking to school to enjoy active activities (e.g., skipping and running) and children going to school with a car to enjoy more passive activities (e.g., listening to music and looking outside the window). Therefore, it is clear that the mode of commuting to school can create different experiences and different interactions with the surrounding physical and social environment. In the case of independent mobility, parents’ decisions define what kind of experiences their children may have along their journey to school. On the other hand, as Ross [22] suggested, independent mobility could enable children to explore their localities. Therefore, designing neighbourhoods and school streets that support no-escort trips could allow children to develop their own experiences of AST without depending on their parents’ social identity. It would be interesting to further explore whether a supportive built environment for AST could shape children’s reflexive motivation for future trips. However, since parental influence is important [44,73] and working parents are more likely to drive children to school [40], it is also recommended that AST behaviour change interventions should, in parallel, target parents’ travel behaviours and their reflexive motivations [55,70].

### 4.2. Study Limitations and Strengths

Our child-centred study had a limited number of participants; however, the random and heterogeneous sampling of schools in different neighbourhoods around the city and the combination of a quantitative sampling technique with a qualitative analysis provide a comprehensive picture of children’s experiences along their journey to/from school. In addition, this is a cross-sectional study, and no causal inferences can be made; therefore, more studies in different spatial and cultural contexts could contribute to a better understanding of children’s affective experiences and their travel behaviours. Although the integration of the behaviour models was part of our results’ interpretations and discussion, to our knowledge, the COM-B model has not been used to explain AST before, and this is the first time the TDF has been used to understand children’s experiences. Hence, it is recommended that future work into the role of the built environment in AST should consider those models as part of their study design, similar to health promotion research from other disciplines [69]. Finally, since the streetscape qualities of positively mentioned places were not surveyed in the current paper, it is suggested that future research could associate the behaviour domains with specific urban forms and design features.

## 5. Conclusions

The results presented in this study contribute to the existing body of knowledge in children’s school travel literature. Focusing on children aged 9–10 in five primary schools located in heterogeneous neighbourhoods in Newcastle upon Tyne, UK, this study identified four primary domains (environmental context, emotions, social influences and trip factors) and specific themes and sub-themes underlying children’s affective experiences of their journeys to school. This study is the first attempt known by the authors to adapt the COM-B model to AST and children’s experiences. The findings indicate that such a multidisciplinary approach holds the potential to promote AST and other healthy behaviours through designing and planning the built environment around schools in an effective way, as a continuous active travel infrastructure. An insight into the principles of behaviour change could inform built environment professionals and researchers and contribute to sustainable behaviour change interventions [29]. Exploring the role of the built environment in AST through children’s affective experiences and behavioural models could help urban planners and designers to understand the underlying reasons for travel behaviours and the environmental impacts on behavioural domains. Engaging the community and relevant stakeholders, such as children, parents and school representatives, during research or pre-intervention process (i.e., through active engagement workshops) could lead to a better understanding of their interaction with the physical environment and, therefore, to more successful built environment interventions [81] aimed to encourage AST. Such an approach could bridge behaviour science and urban forms and enhance the way we plan and design the physical environment. A better understanding of the built environment’s role in human behaviour could lead to a more holistic and multidisciplinary city design, which integrates the concept of health promotion, improving the health of its citizens.

## Figures and Tables

**Figure 1 ijerph-18-04992-f001:**
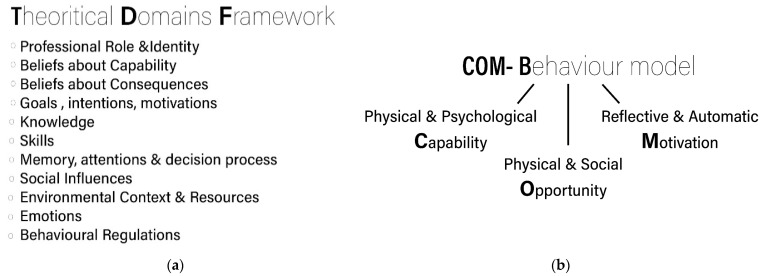
Behaviour change frameworks taking built and social environment into consideration: (**a**) Theoretical Domains Framework (adapted from [54]); (**b**) COM-B model (adapted from [55]).

**Figure 2 ijerph-18-04992-f002:**
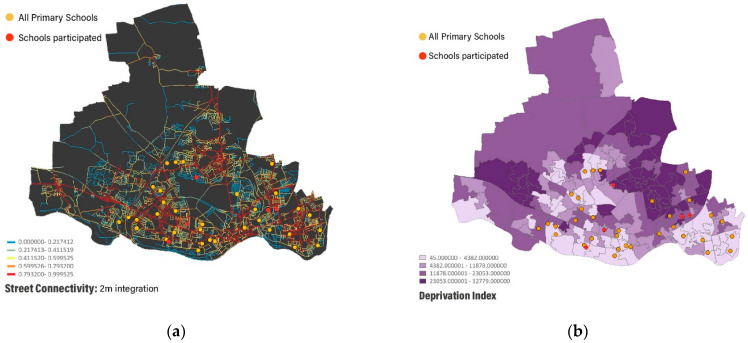
School selective criteria. (**a**) Street connectivity: 2 km integration; (**b**) deprivation rank.

**Figure 3 ijerph-18-04992-f003:**
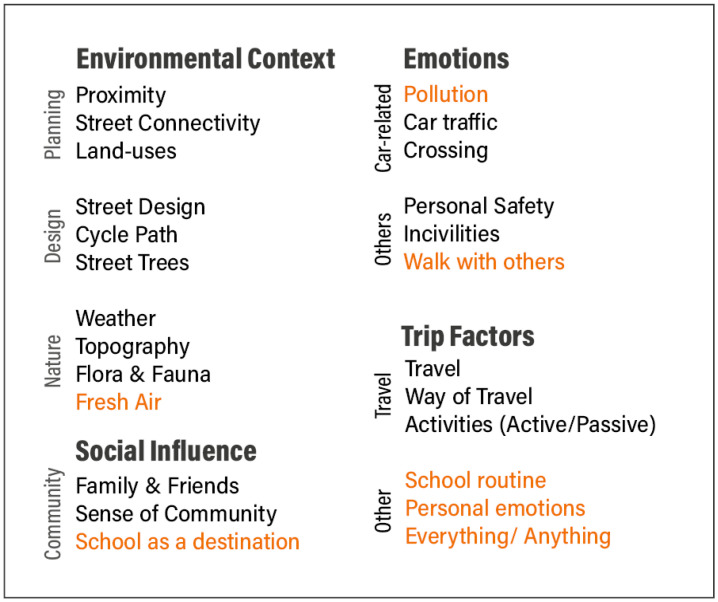
Typologies based on TDF: deductive and inductive (in orange) themes.

**Figure 4 ijerph-18-04992-f004:**
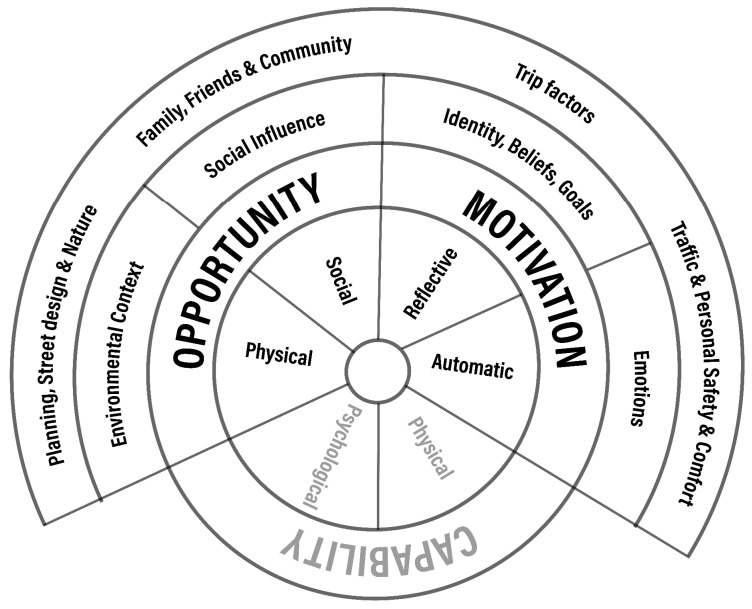
Theoretical domains explored through COM-B (adapted from [55,68]).

**Table 1 ijerph-18-04992-t001:** Participants’ characteristics.

		Walking	Cycling	Car	Park and Stride	Bus	School Bus	NA ^2^	Total
	**Total**	**72**	**7**	**43**	**5**	**4**	**7**	**7**	**145**
**School**	A	31	0	19	0	1	0	6	57
	B	11	0	6	5	0	0	0	22
	C	9	0	3	0	1	1	0	14
	D	19	3	8	0	1	0	1	25
	P ^1^	9	4	7	0	1	6	0	27
**Gender**	Girl	39	2	25	3	2	2	6	79
	Boy	32	5	18	2	2	5	1	65
	NA ^2^	1	0	0	0	0	0	0	1
**Accompanied**	Yes	51	5	36	4	4	1	6	107
	Sometimes	5	0	0	0	0	0	0	5
	No	14	1	0	0	0	2	1	18
	NA ^2^	2	1	7	1	0	4	0	15
**Travel with**	Adult(s)	32	2	17	2	2	1	4	60
	Adult(s) and Child(ren)	18	1	15	1	1	0	1	37
	Child(ren)	6	0	0	0	0	1	1	8
	Alone	6	1	0	0	0	0	0	7
	NA ^2^	10	3	11	2	1	5	1	33
**Stop**	Yes	25	1	21	1	0	1	2	53
	Sometimes	5	0	0	0	0	0	0	5
	No	24	1	8	4	1	0	1	39
	NA ^2^	9	1	7	0	2	0	1	20
	Not Applicable	9	4	7	0	1	6	0	27

^1^ Pilot study school; ^2^ no answer provided.

## Data Availability

The data presented in this study are available on request from the corresponding author. The data are not publicly available due to confidentiality.

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
