# Peer review of "Children’s Experiences of Their Journey to School: Integrating Behaviour Change Frameworks to Inform the Role of the Built Environment in Active School Travel Promotion"

_ijerph, 2021, doi:10.3390/ijerph18094992_

Round 1

Reviewer 1 Report

The present study aimed to detect children’s affective experiences of their journey to school derived from two questionnaires to analyze pupil’s attitudes to promote an  active school travel as opportunity for practicing physical activity and healthy behaviours.

The manuscript is well written and present interesting findings. However, minor modifications are required.

Below are specific comments that might improve the quality of the paper.

Introduction:

Line 46-49: please update the sentence referring to the last published WHO 2020 guidelines

Line 50: please consider to modify “scootering” with another physical activity

Material and Methods

Line 175: please consider to provide a table regarding the mail participants’ features

Discussion

Line 386: please may consider to expand the role of the external context as opportunity to practice physical activity

Reviewer 2 Report

Thank you for your research in this area and such a well written paper. 

I have a suggestion about 2 minor typos - 

Line 15 – is Integration meant to be capitalised?

Line 511 - no need for apostrophe on citizens

My other comment relates to the labelling of the theme around Individual Factors. The other labels for the categories have a meaning that is self-apparent but this is not so for Individual Factors. I appreciate that there are a range of factors involved so it is more difficult to tie them to a label. Individual Factors to me suggested they were related to the individual themselves, whereas on reading the description, it is related to their individual experience of getting there and arriving (or as you describe the circumstances and consequence of the commute). The label is fine as is, but I would like to encourage you to consider whether there is a label that would convey the meaning of the category a bit better.

And my final comment is in relation to the role/relationships between the researchers and the school. The methods described suggest a collaborative relationship with the schools that participated. While the focus of the paper is around influencing design of the built environment, I would like you to consider mentioning the role of the schools and the importance of sharing these findings with schools to inform their efforts towards AST as well (if this was the case). 

Otherwise, thank you again and well done.  
